# The Role of Magnetic Resonance Imaging (MRI) in the Diagnosis of Pancreatic Cystic Lesions (PCLs)

**DOI:** 10.3390/diagnostics13040585

**Published:** 2023-02-05

**Authors:** Elit Quingalahua, Mahmoud M. Al-Hawary, Jorge D. Machicado

**Affiliations:** 1Division of Hematology and Oncology, University of Michigan, Ann Arbor, MI 48109, USA; 2Department of Diagnostic Imaging, University of Texas MD Anderson Cancer Center, Houston, TX 77030, USA; 3Division of Gastroenterology and Hepatology, University of Michigan, Ann Arbor, MI 48109, USA

**Keywords:** magnetic resonance imaging, MRI, pancreas cyst, IPMN, pancreatic cancer

## Abstract

Pancreatic cystic lesions (PCLs) are a common incidental finding on cross-sectional imaging. Given the high signal to noise and contrast resolution, multi-parametric capability and lack of ionizing radiation, magnetic resonance imaging (MRI) has become the non-invasive method of choice to predict cyst type, risk stratify the presence of neoplasia, and monitor changes during surveillance. In many patients with PCLs, the combination of MRI and the patient’s history and demographics will suffice to stratify lesions and guide treatment decisions. In other patients, especially those with worrisome or high-risk features, a multimodal diagnostic approach that includes endoscopic ultrasound (EUS) with fluid analysis, digital pathomics, and/or molecular analysis is often necessary to decide on management options. The application of radiomics and artificial intelligence in MRI may improve the ability to non-invasively stratify PCLs and better guide treatment decisions. This review will summarize the evidence on the evolution of MRI for PCLs, the prevalence of PCLs using MRI, and the MRI features to diagnose specific PCL types and early malignancy. We will also describe topics such as the utility of gadolinium and secretin in MRIs of PCLs, the limitations of MRI for PCLs, and future directions.

## 1. Introduction

Pancreatic cystic lesions (PCLs) are often found when performing cross-sectional imaging for unrelated reasons. Magnetic resonance imaging (MRI) is considered the standard non-invasive imaging modality for detecting and evaluating PCLs [1,2]. Compared with computed tomography (CT) or transabdominal ultrasound, MRI with magnetic resonance cholangiopancreatography (MRCP) is more sensitive to detect small PCLs and provides superior contrast resolution to delineate cyst septations, nodules, or duct communication [3,4]. As PCLs often require surveillance imaging, MRI also has the advantage of lacking ionizing radiation and avoiding potentially nephrotoxic iodine-contrast required for CT in patients with significant renal dysfunction.

Over the past few decades, the image quality and spatial resolution of MRI have improved due to advances in MRI software and hardware. These improvements have resulted in increased detection of PCLs, impacting the epidemiology of these lesions [5]. In addition, MRI has become crucial in non-invasively characterizing PCLs, predicting cyst histology, risk stratifying the presence of neoplasia in mucinous cysts, and monitoring changes to PCLs during surveillance. More recently, the use of radiomics and artificial intelligence (AI) seems promising and may expand the critical role of MRI in the evaluation of PCLs. In this review, we will discuss all these aspects and summarize our current knowledge on the application of MRI in the detection and diagnosis of PCLs.

## 2. Literature Search

One of the authors (E.Q.) conducted the electronic search. We searched MEDLINE via PubMed from inception through November 2022 to identify relevant studies. The search was limited to studies published using the English language. The electronic search strategy was conducted using a combination of phrases indicating the diseases of interest [“pancreatic cyst”, “pancreatic cystic lesion(s)”, “intraductal papillary mucinous neoplasm”, “mucinous cyst”] and imaging technology [“magnetic resonance imaging”, “MRI”, “MRCP”]. To identify additional potential studies, we reviewed the reference lists of the eligible primary studies and reviews.

## 3. Evolution of MRI on Assessment of PCLs

MRI has been used in the evaluation of PCLs for almost four decades [6]. Initial studies reporting the utility of MRI in PCLs were small and were done with low-field magnets strength of 0.35 Tesla (T, unit that measures the strength of the magnet) [7]. The most commonly current magnet strengths in clinical use today are 1.5 and 3 T. These systems have an improved signal-to-noise ratio and higher spatial resolution imaging, although at a cost of higher susceptibility to image degradation by metal and air [5]. The addition of MRCP sequences to MRI started three decades ago, initially with 2D techniques and more recently with 3D sequences that enable near isotropic image acquisition that offers the highest image resolution [8,9]. Additional improvements also lead to a decrease in scan time acquisition and introduction of new sequences that can provide additional information (e.g., diffusion-weighted imaging [DWI]]) [10,11].

A recent statement from the Society of Abdominal Radiology suggested that the assessment of PCLs should be performed with MRIs of at least 1.5 T magnets equipped with phased-array coils [2]. Although MRI protocols are variable across institutions, a conventional MRI protocol for the initial assessment of PCLs should include: (A) T1-weighted images: useful in assessing fat or blood content and in assessing presence of enhancement when compared to the post contrast images; (B) T1-weighted postcontrast image: obtained after gadolinium is administered, shows low/delayed enhancement with adenocarcinoma and generally early enhancement with neuroendocrine tumor; (C) T2-weighted images: useful in assessing fluid containing structures; (D) MRCP: relies heavily on T-2 weighted sequence, suppresses non-fluid signals from surrounding parenchyma to accentuate ductal anatomy; (E) DWI: increasingly used in PCLs, as it helps identifying small mural nodules and early cancer [2,11].

## 4. Prevalence of PCLs in MRI

In a systematic review of 17 observational studies, the prevalence of PCLs incidentally found in cross-sectional images was 8% (95% CI, 4–14) [12]. The prevalence of PCLs was higher in studies that employed MRCP as compared to those that used CT, confirming higher detection of PCLs with MRI. Table 1 portraits a summary of the studies that have evaluated the prevalence of incidental PCLs with MRI. The large variations in PCL prevalence estimates across observational studies and countries may be explained by differences in MRI hardware/software, quality of radiologic interpretation, or ethnic/genetic predisposition. The effect of technological improvements in PCL prevalence was illustrated in a study of 500 patients from Mayo Clinic, that found a strong association between PCL detection and improvements on both MRI software and hardware [5]. This study also showed that the strength of the magnet was not associated with the detection of PCLs.

## 5. MRI Features of PCLs

The key features of PCLs should be reported in a standardized manner to guide clinicians in the management of these patients and to facilitate multidisciplinary discussion. The number (single vs. multifocal), location, and dimensions of PCLs should be reported. Previous studies have found significant variability in measuring cysts. In a study of 144 PCLs, there was overall excellent interobserver agreement on measuring absolute cyst size (k = 0.81) among four radiologists, but with variability of 4 mm between readings and with only moderate interobserver agreement (k = 0.59) when assigning a size category (<10, 10–19, 20–29, ≥30 mm) [24]. Since the cyst dimensions have important implications in treatment decisions, there is a need for reproducible and consistent methods to measure PCLs. A recommended approach is measuring the longest diameter from outer-wall to outer-wall in any plane that demonstrates the largest measurement (axial, coronal, or sagittal plane) using 2D or 3D MRCP sequence [2]. Although differentiating a single multiseptated cyst from adjacent clustered PCLs is difficult, the entire cluster should be measured as a single lesion. The images and series number should be documented in the report and the measurements should be saved in the imaging archive, so the information can be used in subsequent follow-ups. The cyst growth rate should also be mentioned in the report when comparing to prior MRIs.

Morphologically, pancreatic cysts can be unilocular or multilocular. When multilocular, they can be macrocystic (individual compartment > 2 cm) or microcystic (contains multiple small components of <2 cm). The shape is variable, ranging from round, oval, pleomorphic, or with the appearance of a “bunch of grapes”. Some cysts have calcifications, which can be central, peripheral, or distributed as a circumferential rim. The presence and size of mural nodules is very important as a marker of advanced neoplasia. Thickening of the cyst wall (>2 mm) or thickened septations (>2 mm) should be reported when present. When contrast is administered, enhancement of the wall, septation, and/or mural nodule should also be documented.

In addition to characterizing the cyst, MRI interpretation should thoroughly assess the pancreatic duct, the pancreatic parenchyma, and the peripancreatic soft tissue. The pancreatic duct should be measured perpendicular to its long axis, ideally in coronal or axial T2-weighted or MRCP images. The threshold to define ductal dilation is 5 mm in the head, 4 mm in neck/body, and 3 mm in the tail [2]. Ductal dilation can be diffuse, involving the entire main pancreatic duct; or segmental, with only a portion of the pancreatic duct being dilated with a smooth transition to a normal caliber duct. Segmental dilation can occur downstream the cyst, suggesting mucin over-production or mixed duct intraductal papillary mucinous neoplasm (IPMN); or can occur upstream the cyst, which should raise concern for stricture secondary to malignant transformation. Ductal communication of the cyst should also be specified.

## 6. Role of MRI to Diagnose Specific Cyst Types

MRI plays a pivotal role in the diagnostic work-up of PCLs. Most patients with PCLs do not undergo endoscopic ultrasound or surgical resection, so cyst diagnosis in clinical practice heavily relies on proper imaging interpretation, demographic characteristics, and clinical features. Most cysts are mucinous (IPMNs and mucinous cystic neoplasms [MCNs]) and carry the potential to transform into invasive cancer and pancreatic ductal adenocarcinoma (PDAC), which is on the rise to become the second leading cause of cancer-related deaths in the United States. Although malignant transformation is uncommon and only one-sixth of all PDACs arise from a mucinous cyst, their identification offers an opportunity for surveillance and early detection of PDAC [25,26]. Non-mucinous cysts are less common and include a diversity of PCLs with variable malignant potential: (A) Benign: serous cystadenoma (SCA), pseudocyst, retention cyst, lymphoepithelial cyst and (B) Malignant: cystic-neuroendocrine tumor (NET), solid pseudopapillary neoplasm (SPN) and rarely cystic adenocarcinoma. Accurate diagnosis of specific cyst type offers a pathway for precision medicine of PCLs, by preventing unnecessary surgeries or surveillance tests for benign cysts, monitoring pre-malignant cysts without malignant transformation, and surgically resecting malignant cysts if feasible. In this section, we will summarize the role of MRI in differentiating among the most common PCLs.

### 6.1. Intraductal Papillary Mucinous Neoplasm (IPMN)

IPMNs are the most common type of PCLs. These are pre-malignant lesions arising from the ductal epithelium and are composed of mucin-producing columnar cells. The incidence of IPMNs increases with age, especially by the seventh decade of life [27]. IPMNs can be classified as: branch-duct (BD), main-duct (MD), or mixed-type (both). The risk of malignant transformation is greater among main-duct and mixed-duct IPMNs as compared to branch duct-IPMN [1].

The landmark feature of branch-duct IPMNs is the direct communication of the cyst with the main pancreatic duct. In a study of 24 IPMNs with surgical histopathology, radiologists identified ductal communication of PCLs with MRCP in 21/24 lesions with 87% sensitivity [28]. In another study of 53 patients with PCLs (31 IPMNs), MRCP established ductal communication with over 90% accuracy (area under the curve [AUC] 0.92–0.95) [29]. Similar findings were reported in a study of 34 patients with branch-duct IPMNs that found that 3D MRCP was highly accurate to predict ductal communication (AUC 0.96) [30]. However, ductal communication is not noticeable in all branch-duct IPMNs, and can potentially lead to false positive results as it may also be present in pseudocysts (which are essentially contained pancreatic duct leaks). Sometimes it can also be challenging to differentiate a dilated side-branch (as seen in chronic pancreatitis, for example) from a small branch-duct IPMN, but a size of >5 mm should indicate IPMN.

Branch-duct IPMNs can be single or multifocal (Figure 1). Multifocal lesions have been reported in 14–41% of patients with branch-duct IPMNs, often localized to one segment of the pancreas, and sometimes distributed diffusely [2,31]. The risk of advanced neoplasia has been found to be similar between single and multifocal IPMNs [2,31]. Morphologically, branch-duct IPMNs can have different patterns: (A) unilocular; (B) multilocular with macrocystic pattern and few septations; (C) microcystic pattern with a grape-fruit appearance, in which multiple septations separate numerous fluid spaces; (D) pleomorphic, with a combination of these patterns [32]. None of these morphologic features are specific of IPMNs, but can help narrow the differential diagnosis. The ability of these lesions to produce mucin can sometimes lead to mucin plugs or balls within the cyst or in the pancreatic duct. These often present as non-enhancing mural nodules on the dependent surface of the cyst and should not be confused with solid component.

Given the involvement of the pancreatic duct, main-duct and mixed-duct IPMNs are characterized by pancreatic ductal dilation (Figure 2). However, the main pancreatic duct can also be dilated in the setting of branch-duct IPMN due to mucin over-production but usually will not be more than 1 cm, which is more specific for main duct involvement. The differential diagnosis for isolated pancreatic ductal dilation is even broader, including pancreatic cancer, ampullary malignancy, age-related changes, papillary stenosis, pancreatic ductal stricture, and chronic pancreatitis. Moreover, main-duct IPMN may co-exist with chronic pancreatitis and have typical findings of advanced chronic pancreatitis (e.g., calcifications, intraductal stones) [33].

### 6.2. Mucinous Cystic Neoplasm (MCN)

MCNs are mucin-producing cysts with a characteristic ovarian-type stroma, for which they almost exclusively occur in women. These lesions develop more often between the fifth and sixth decade of life and are mainly located in the body or tail of the pancreas [34]. MCNs have malignant potential, but the risk of malignant transformation is small [35].

In MRI images, MCNs often present as a single unilocular cyst without communication to the pancreatic duct (Figure 3). They can also have a multilocular, macrocystic pattern with thin septations [27]. The cyst wall is often thick and can enhance in delayed contrast-enhanced MRI, which correlates with fibrotic changes noticed in histology [36]. A subset of these lesions (10–25%) have peripheral or eggshell calcification which are better assessed on CT (Figure 4) [32]. The content of the cyst is usually homogeneous, with low T1-weighted signal intensity and high T2-weighted signal intensity. Although all these MRI features are helpful, the diagnosis of a MCN usually requires consideration of the demographic features and a multimodal diagnostic approach.

### 6.3. Serous Cystic Adenoma (SCA)

SCAs are benign lesions that occur slightly more common in women, between the fifth and seven decades of life [37]. These lesions can develop anywhere in the pancreas and grow approximately 3 mm per year [38]. Patients with Von Hippel-Lindau (VHL) are at increased risk of developing SCAs [39].

The majority of SCAs (70–80%) have a typical microcystic appearance, which on MRI is seen as a cluster of small T2 hyperintense cystic lesions [40]. The outer margin of these lesions is regular, with a lobulated contour, and with a thin wall (Figure 5). Microcystic SCAs have thin enhancing septations that are highly vascular and can sometimes bleed. As these cysts grow, the fibrous tissue can retract, and this yields a central calcified stellate scar. This occurs in 20–30% of cases, especially in large lesions (>5 cm), and is characteristic of SCAs. In 20% of SCAs, the cyst can have a “honeycomb” pattern, in which innumerable tiny T2 hyperintense cysts are clustered [41].

A multilocular, macrocystic pattern is noticed in <10% of SCAs [40]. This variant presents with fewer and larger cystic components (2–7 cm), tends to be lobulated, located in the head of the pancreas, and has thin septations (Figure 6). Other patterns are rare but have been reported: (A) unilocular SCA; (B) giant SCA (>10 cm), which may compress surrounding structures and cause ductal dilation; (C) solid-appearing SCA, which has microscopic cysts that are too small to be detected on MRI and appears solid; (D) disseminated SCA, which can be seen with VHL [40].

### 6.4. Pseudocyst

Pancreatic pseudocysts are fluid collections surrounded by a non-epithelialized wall made up of fibrous and granulation tissue. The diagnosis of a pseudocyst is aided by a well-documented history of acute pancreatitis or radiologic evidence of chronic pancreatitis (e.g., calcifications, intraductal stones). However, its diagnosis is sometimes difficult when there is no history of acute pancreatitis or radiologic findings definitive of chronic pancreatitis. Moreover, when the cyst is found in the setting of acute pancreatitis and no prior imaging is available, it may be challenging to confidently determine if the lesion is a pre-existing mucinous cyst causing pancreatitis or an acquired inflammatory collection caused by pancreatitis (Figure 7).

On MRI, pseudocysts are typically unilocular, with round or oval morphology, and with a well-defined fibrous wall. They often contain homogenous fluid intensity and no internal septations. However, none of these imaging features is specific and can also be found in pre-malignant cysts. The presence of calcifications and parenchymal atrophy favors the diagnosis of a pseudocyst. Sometimes layering non-enhancing debris due to blood products or necrotic material is noticed in the evaluation of pseudocysts with MRI, especially on T2-weighted sequences. In a retrospective single-center study of 42 patients with PCLs (20 pseudocysts and 22 other cystic neoplasms), detection of debris was highly specific (95%) and had high positive predictive value (92%), but with limited sensitivity (60%) [42]. Experienced abdominal radiologists had almost perfect inter-observer agreement (K = 0.89) in identifying this radiologic finding. However, not all pseudocysts have debris on MRI. Some pseudocysts may have internal septations, a microcystic pattern, or even ductal communication, which can resemble SCAs or branch-duct IPMNs.

### 6.5. Lymphoepithelial Cysts

Lymphoepithelial cysts are rare benign PCLs, composed of squamous epithelial cells and lymphoid tissue. These lesions occur predominantly in men in their sixth decade of life. On MRI, they appear multilocular or unilocular [43]. Due to its keratinized content, these cysts can have high signal intensity on T1 weighted images, low signal intensity on T2 weighted images, and profound water restriction in DWI [44]. None of these radiologic features are specific for a definitive pre-operative diagnosis and endoscopic ultrasound (EUS) is often needed [44]. The presence of squamous cells with lymphocytes on cytological examination is diagnostic of lymphoepithelial cyst [45].

### 6.6. Cystic Neuroendocrine Tumor (Cystic-NET)

Cystic NETs are rare PCLs, that have no gender predilection, and present often in the sixth decade of life [32]. Cystic changes are present in 17% of all resected NETs [46]. This is believed to occur secondary to tumor degeneration and is localized in the central portion of the lesion. These tumors are more commonly located in the body and tail of the pancreas. Although pancreatic NETs are difficult to diagnose radiographically, MRI can show a relatively thick wall and rim enhancement during the post-contrast arterial phase that would differentiate them from the other cystic lesions [32].

### 6.7. Solid Pseudopapillary Neoplasm (SPN)

SPNs of the pancreas are rare neoplasms that occur more often in females during the third decade of life [47]. The cystic component in SPN is due to tumor degeneration. On MRI, SPNs present as large, solitary, encapsulated lesions with a heterogeneous T2- weighted signal due to the solid and cystic component (Figure 8) [48]. These lesions have an oval or lobulated shape, contain blood (which is hyperintense on T1-weighted sequences) and debris, and are often located in the pancreatic body or tail [32]. SPNs have slow progressive enhancement after contrast administration, which helps to differentiate them from other lesions [49].

### 6.8. Retention Cysts

Pancreatic retention cysts are rare benign lesions that can be found in the setting of chronic pancreatitis, cystic fibrosis, or pancreatic tumors [50]. Mechanistically, these cysts represent a dilated segment of the pancreatic duct due to focal duct obstruction from strictures, mucin plugs, stones, or tumors. The radiologic appearance of retention cysts has not been widely described and it may be difficult to diagnose these cysts with MRI alone as they share characteristics with other PCLs. In a small study of 16 patients with retention cysts, these lesions were well-defined, round, without solid component hypointense on T_1_WI images, and hyperintense on T_2_WI images [50]. These lesions may have thin septations, communication to the main pancreatic duct, and ductal dilation.

## 7. Role of MRI in Detecting Advanced Neoplasia

Once a PCL is characterized as mucinous, the next step is determining if advanced neoplasia is present. Several society guidelines have proposed classification systems to risk stratify the malignant potential of mucinous lesions based on the presence of high-risk or worrisome imaging features [1,51,52,53,54]. Although the grading parameters are variable across guidelines, all societies agree that the presence of an enhancing or solid mural nodule, pancreatic ductal dilation, and jaundice, represent high-risk features concerning for malignancy and should prompt consideration for surgery. The radiologic thresholds to define these high-risk features are variable. The European and Fukuoka guidelines require that the mural nodule is ≥5 mm and enhancing in T1-weighted postcontrast images to define it as a high-risk feature [1,51,52]; while the American Gastroenterology Association (AGA) and American College of Gastroenterology (ACG) guidelines use less specific definitions [53,54]. The finding of a mural nodule on its own is the strongest radiologic predictor of neoplasia in PCLs (sensitivity 48%, specificality 91%) and has shown to be associated with 6–7 greater risk of harboring malignancy as compared to those without it [26,55]. With regards to pancreatic duct diameter, the AGA and ACG guidelines are less strict, requiring a diameter of at least 5 mm to define a high-risk feature [53,54]; while the American College of Radiologist (ACR), European and Fukuoka guidelines require a diameter of ≥10 mm [1,51,52]. A cutoff of 5 mm provides 56% sensitivity and 67% specificity to predict malignancy [55], while a cutoff of 10 mm results in high specificity of 98% but low sensitivity of 13% [56]. Other features that are worrisome and less predictive of malignancy include: cyst size ≥3 cm, enhancing cyst wall, abrupt caliber change of the duct, lymphadenopathy, and cyst growth ≥5 mm in 2 years [2]. When present, these features often lead to endoscopic ultrasound, multidisciplinary discussion, and decision about surgery or more frequent surveillance based on individual factors. If high-risk and worrisome features are absent, imaging surveillance should be offered if the patient remains surgically fit and willing to undergo surgical resection if needed.

## 8. Methods to Improve Evaluation of PCLs with MRI

### 8.1. Use of Secretin

Secretin stimulates bicarbonate-rich fluid secretion in the pancreas, causing a temporary dilation of the pancreatic duct and improved ductal visualization. In theory, secretin may potentially uncover small ductal communication of PCLs, assisting in the differentiation of branch-duct IPMNs vs. other PCLs. However, data supporting its role in PCLs are still equivocal [57]. In a prospective multicenter study of 64 subjects with small PCLs (average size 7 mm), the addition of secretin increased the detection of ductal communication by 4.7% as compared to MRCP pre-secretin (54.7% vs. 50%, OR= 1.28, *p* = 0.04) and improved the diagnostic confidence of reading radiologists [58]. In a more recent prospective study of 21 patients with PCLs (median size 18 mm), the addition of secretin did not significantly increase ductal communication (published as abstract) [59]. When used, the recommended dose of intravenous secretin is 0.2 ug/kg (16 ug in a 80-kg subject) and its peak action is at 3–5 min. The medication is safe, causing mild adverse events in 0.5% of patients (e.g., nausea, abdominal pain, flushing) [57]. However, the addition of secretin increases costs, adds at least 15 min to the exam, and requires a nurse to administer the intravenous infusion. Hence, and given its unproven benefits for PCLs, its routine use for PCLs is not justified.

### 8.2. Use of Gadolinium

Gadolinium-based contrast media is used to obtain T1-weighted postcontrast images. This can detect and characterize mural nodules within PCLs and identify subtle metachronous solid lesions [60]. In current guidelines, an enhancing mural nodule is considered a high-risk feature, while a non-enhancing nodule is considered at the most, a worrisome feature [1,51]. Thus, contrast-enhanced MRI should be performed on initial characterization of PCLs for risk stratification of advanced neoplasia [2]. However, the benefits of intravenous contrast on follow-up MRIs of PCLs has been put into question. In a study of 56 patients with PCLs, two radiologists had substantial intra-observer reliability (k = 0.67) on the interpretation of surveillance MRIs with or without gadolinium-enhanced images [60]. When they evaluated the disagreements in more detail, investigators found that there was nothing on the gadolinium-enhanced sequences that would specifically alter treatment decisions. In a more recent study of 87 PCLs (11 with advanced neoplasia), the addition of gadolinium did not predict advanced neoplasia more accurately than unenhanced MRIs [61]. Other retrospective studies have also found that an abbreviated MRI protocol without contrast would suffice for surveillance of known PCLs without implications in management [62,63]. Surveillance imaging with contrast-enhanced MRI is likely justified in patients with worrisome and/or high-risk features, suspicious symptoms, family history of PDAC, genetic predisposition, or prior surgical resection of advanced neoplasia [2].

### 8.3. Radiomics

Radiomics, also known as quantitative imaging or texture analysis, involves extracting numerous features of images and converting them into mathematical models. Until recently, most studies had applied radiomics to CTs of PCLs with promising results [64]. However, emerging research is evaluating the role of radiomics in MRIs of PCLs. A recent study of 59 subjects with a variety of PCLs showed that differentiation of mucinous from non-mucinous lesions was made with AUC 0.70–0.97 using T2-based radiomic features and AUC 0.74–0.87 using apparent diffusion coefficient-based features [65]. Moreover, in another study of 248 patients with IPMNs (106 with high grade dysplasia or cancer), the addition of radiomic features to traditional MRI interpretation increased the accuracy to detect early neoplasia from 0.8 to 0.85 [66]. Future studies are required to better understand the role of radiomics in the evaluation of PCLs.

### 8.4. Artificial Intelligence (AI)

AI can have a tremendous impact in the field. Deep learning algorithms can play a major role in image acquisition, reconstruction, and image post-processing. Moreover, AI carries the potential to automate the diagnosis and risk stratification of PCLs using cross-sectional images, endoscopic ultrasound, digital pathomics, and genomics [67]. In a study of 139 patients with PCLs, the integration of a deep learning algorithm to MRI classified advanced neoplasia with 75% sensitivity and 78% specificity (AUC 0.783) [68]. However, the accuracy of this AI model was comparable to current management guidelines, so further iterations and more sophisticated models are needed. AI also carries the potential to automate the redaction of MRI reports and the determination of surveillance intervals.

## 9. Limitations of MRI

Despite advances in MRI technology, accurate diagnosis of specific cyst types and determination of the degree of neoplasia remains a major challenge with imaging alone. This is due to overlapping, non-specific MRI findings from the different PCLs, inter-observer variation on imaging interpretation, and variability in the type of MRI scanner used. Another limitation with MRI is the duration of the test, which takes 30 to 45 min, is longer than other imaging modalities, and can result in motion artifacts. This, coupled with the enclosing nature of the scanning equipment, scanning noise, and lying flat position, are factors that contribute to claustrophobia, which occurs in 1–10% of all patients undergoing MRI [69]. Claustrophobia can sometimes be mitigated with sedation or open MRI scanners. However, despite these interventions, ~1% of patients do not tolerate MRI due to claustrophobia and require an alternative imaging modality [70]. Given the use of magnetic fields, there are several potential contraindications in patients with metallic implants (e.g., pacemaker, implanted cardioverter-defibrillator, joint replacement) that need to be carefully reviewed prior to MRI [71]. Recent technological developments have led to the reduction in the use of ferromagnetic material inside many metallic implants and have resulted in the increased proportion of MRI compatible devices. Finally, in patients who are critically ill, who cannot lie flat, or who cannot hold their breath, MRI for PCL evaluation should be deferred until medical optimization.

## 10. Conclusions and Future Directions

The future of MRI in diagnosing PCLs is bright. There are several clinical needs in the field, including shortening scan times, optimizing imaging quality, automating analyses/reports, and automating surveillance intervals. In addition, as many patients with PCLs are diagnosed and managed at non-academic centers, there is widespread need at the community level for high-quality MRI scans, trained staff conducting the imaging protocols, and specialized abdominal radiologists interpreting these images. In most PCLs, MRI and the patient’s history will suffice to stratify lesions and surveillance needs. The discovery of a non-invasive blood biomarker that can accurately diagnose PCLs is needed to supplement the information provided by MRI. Despite the anticipated progress, MRI alone will unlikely be sufficient in all patients with PCLs, and some patients will need a multimodal approach (e.g., EUS with fluid analysis [72], confocal endomicroscopy [73], molecular analysis [74]) to accurately diagnose the type of PCL and determine the degree of neoplasia.

## Figures and Tables

**Figure 1 diagnostics-13-00585-f001:**
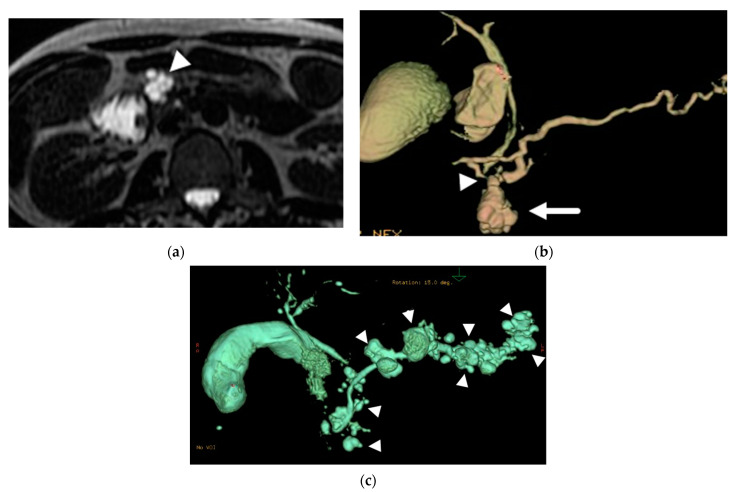
Branch duct IPMN. (**a**) Axial T2-weighted image demonstrates a multilocular cystic lesion in the pancreatic head (arrowhead). (**b**) Volume rendered image from MRCP demonstrates the bunch of grape appearance (arrow) with duct communication (arrowhead), confirming the diagnosis of BD-IPMN. (**c**) Volume rendered MRCP from another patient demonstrates numerous branch duct IPMNs throughout the gland (arrowheads).

**Figure 2 diagnostics-13-00585-f002:**
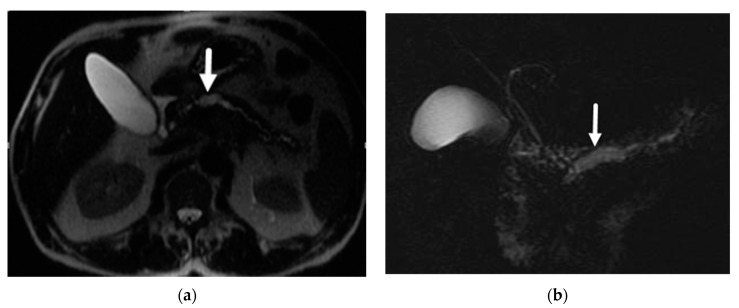
Main duct IPMN. (**a**) Axial T2-wieghted image demonstrates focal dilatation of the PD in the body (arrow). (**b**) MRCP demonstrates the focal dilatation in relation to the rest of the duct.

**Figure 3 diagnostics-13-00585-f003:**
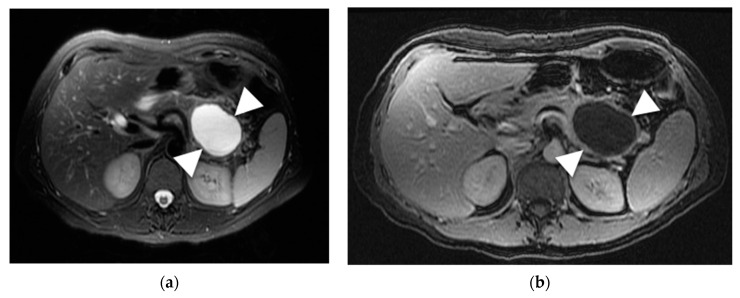
MCN. Middle-aged female with incidental pancreatic lesion. (**a**) Axial T2-weighted images demonstrate a pancreatic tail lesion (arrowheads) with high signal on the T2-weighted image, indicating fluid content and cystic nature. (**b**) T1-weighted post IV contrast image shows low signal centrally, indicating lack of solid enhancing component.

**Figure 4 diagnostics-13-00585-f004:**
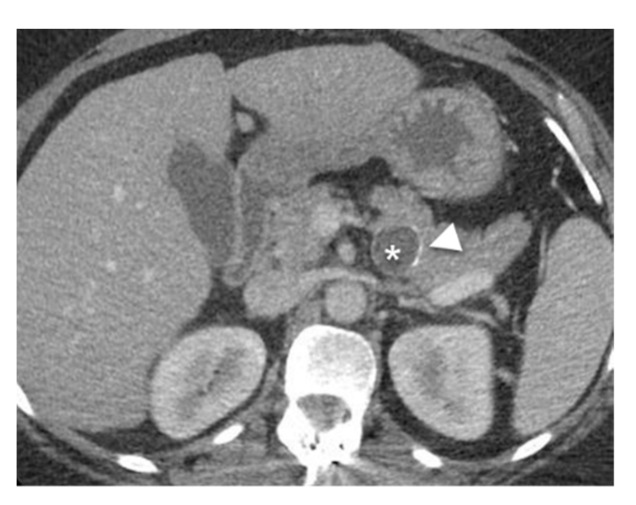
MCN. Axial CT image with IV contrast demonstrates a round cystic lesion in the pancreatic body (asterisk) with rim calcification (arrowhead).

**Figure 5 diagnostics-13-00585-f005:**
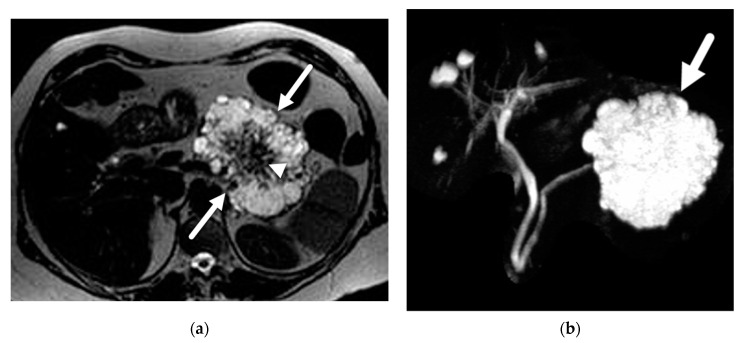
SCA. (**a**) Axial T2-weighted image demonstrates a hyperintense microcystic lesion in the pancreatic tail (arrows) with central scar (arrowhead). (**b**) MRCP demonstrates lobular contour of the lesion (arrow). (**c**) Axial CT post IV contrast shows the sun burst calcification in the central scar (arrowhead).

**Figure 6 diagnostics-13-00585-f006:**
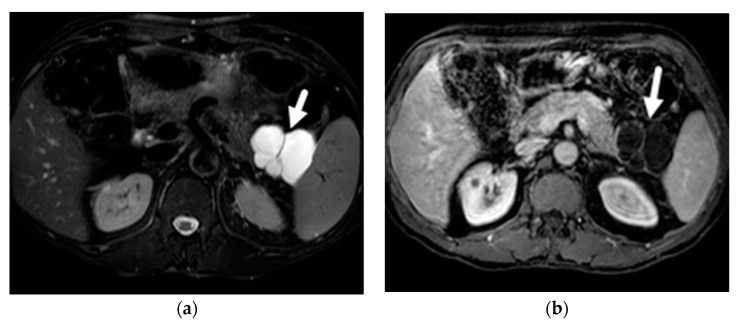
Macrocytic SCA. (**a**) Axial T2-weighted image demonstrates a macrocystic lesion with few septations in the pancreatic tail (arrow). (**b**) T1-weighted post contrast image demonstrates the thin enhancing septa. The lesion represented a SCA on surgical pathology.

**Figure 7 diagnostics-13-00585-f007:**
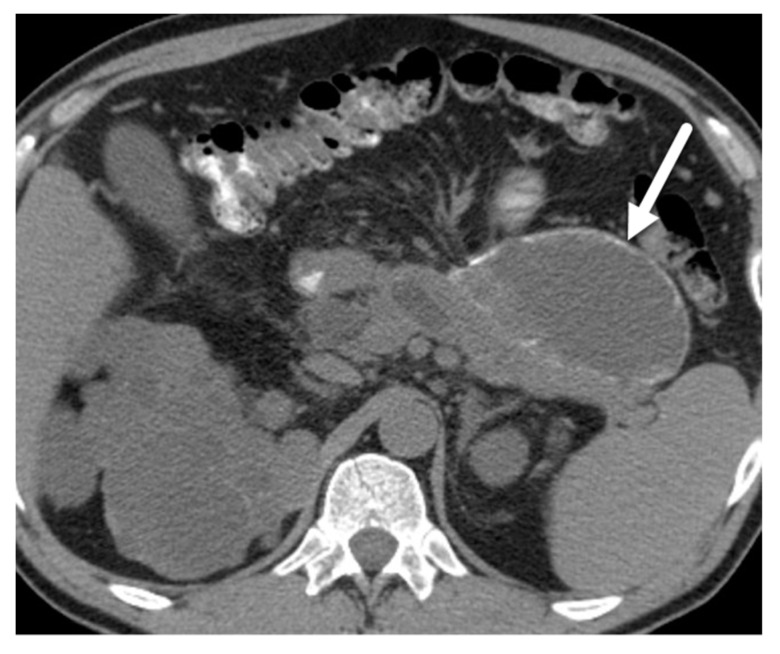
Pseudocyst. Axial CT without IV contrast demonstrates a cystic lesion with rim calcification in the pancreatic tail (arrow). Cyst analysis confirmed a pseudocyst.

**Figure 8 diagnostics-13-00585-f008:**
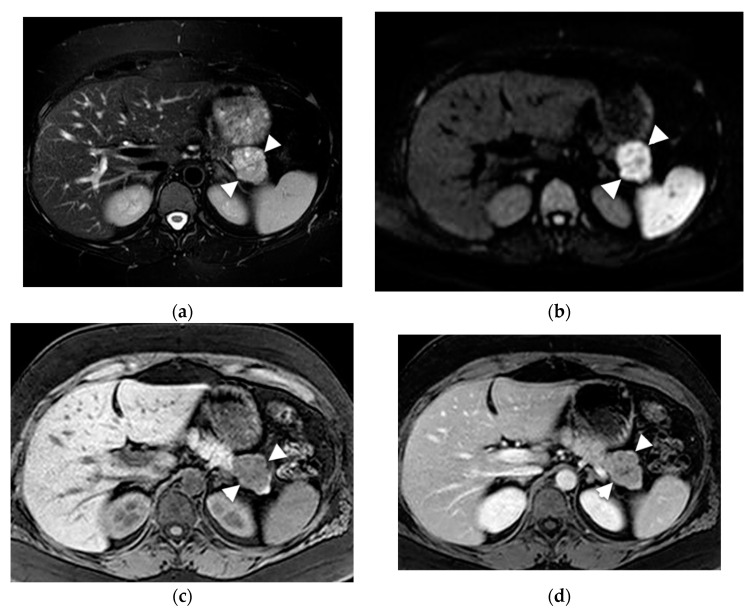
SPN. Young female with incidental pancreatic lesion. (**a**) Axial T2-weighted image demonstrates a heterogenous T2 hyperintense lesion in the tail (arrowheads). (**b**) Axial DWI image demonstrates increased signal indicating a cellular lesion. (**c**) Axial T1-weighted pre and (**d**) T1-weighted post IV contrast indicating central enhancement in the lesion.

**Table 1 diagnostics-13-00585-t001:** Prevalence of incidentally detected PCLs in observational studies.

Author	Year	Country	Design	Magnet	N	PCLs	Prevalence
De Jong [13]	2010	Holland	Retrospective	1.5 T, MRI	2803	66	2.4%
Lee [14]	2010	USA	Retrospective	1.5 T, MRI	616	83	13.5%
Girometti [15]	2011	Italy	Retrospective	1.5 T MRI/MRCP	152	68	44.7%
Matsubara [16]	2012	Japan	Retrospective	1.5 T, MRI/MRCP	1226	123	10%
De Oliveira [17]	2015	Brazil	Retrospective	3 T, MRI	2583	239	9.3%
Ulus [18]	2016	Turkey	Prospective	1.5 T, MRI	118	1	0.8%
Moris [5]	2016	USA	Retrospective	1.5T/3 T, MRI/MRCP	500	208	41.6%
Kim [19]	2016	USA	Retrospective	1.5 T, MRI	110	25	22.7%
Mizuno [20]	2017	Japan	Retrospective	3 T, MRI/MRCP	5296	724	13.7%
Kromrey [21]	2018	Germany	Retrospective	1.5 T, MRI/MRCP	1077	494	45.9%
Zhu [22]	2019	China	Retrospective	1.5 T, MRI/MRCP	38,099	1282	3.4%
Sun [23]	2019	China	Retrospective	1.5 T/3 T, MRI	10,987	212	1.9%

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
