# Peer review of "The Role of Magnetic Resonance Imaging (MRI) in the Diagnosis of Pancreatic Cystic Lesions (PCLs)"

_diagnostics, 2023, doi:10.3390/diagnostics13040585_

Round 1
Reviewer 1 Report
This manuscript title: The role of magnetic resonance imaging (MRI) in the diagnosis of pancreatic cystic lesions (PCLs), makes a revision about the MRI as a diagnostic method.
It is well written and offer an excellent overview about different options that MRI offers to explore the healthiness of the pancreas. However, there are some considerations to check:
1-It is very important to know which different motor of searching have been used. It is not methodology explained to check. A methodology should be included according PRISMA protocol:
METHODS
Eligibility criteria 8 Specify the study characteristics (such as PICO, study design, setting, time frame) and report characteristics (such as years
considered, language, publication status) to be used as criteria for eligibility for the review
Information sources 9 Describe all intended information sources (such as electronic databases, contact with study authors, trial registers or other
grey literature sources) with planned dates of coverage
Search strategy 10 Present draft of search strategy to be used for at least one electronic database, including planned limits, such that it could be
repeated
* It is strongly recommended that this checklist be read in conjunction with the PRISMA-P Explanation and Elaboration (cite when available) for important clarification on the items. Amendments to a review protocol should be tracked and dated. The copyright for PRISMA-P (including checklist) is held by the PRISMA-P Group and is distributed under a Creative Commons Attribution Licence 4.0.
2-2019 is the last year checked, I suppose because there is not methods to know if this is true. That is the reason to ad a good methods section in a systematic review even if it is only a simple review.
One the methods section will be implemented could be consider to publication.

Author Response
We appreciate the comments from the reviewers. We have responded to each of the queries. Please find our responses in red.
Reviewer 1
This manuscript title: The role of magnetic resonance imaging (MRI) in the diagnosis of pancreatic cystic lesions (PCLs), makes a revision about the MRI as a diagnostic method.
It is well written and offer an excellent overview about different options that MRI offers to explore the healthiness of the pancreas. However, there are some considerations to check:
1-It is very important to know which different motor of searching have been used. It is not methodology explained to check. A methodology should be included according PRISMA protocol:
METHODS
Eligibility criteria 8 Specify the study characteristics (such as PICO, study design, setting, time frame) and report characteristics (such as years
considered, language, publication status) to be used as criteria for eligibility for the review
Information sources 9 Describe all intended information sources (such as electronic databases, contact with study authors, trial registers or other
grey literature sources) with planned dates of coverage
Search strategy 10 Present draft of search strategy to be used for at least one electronic database, including planned limits, such that it could be
repeated
* It is strongly recommended that this checklist be read in conjunction with the PRISMA-P Explanation and Elaboration (cite when available) for important clarification on the items. Amendments to a review protocol should be tracked and dated. The copyright for PRISMA-P (including checklist) is held by the PRISMA-P Group and is distributed under a Creative Commons Attribution Licence 4.0.
2-2019 is the last year checked, I suppose because there is not methods to know if this is true. That is the reason to ad a good methods section in a systematic review even if it is only a simple review.
One the methods section will be implemented could be consider to publication.
Thank you for the comments and suggestions. A section in how the literature search was conducted has been added to the manuscript (section 3).
Reviewer 2 Report
Language and grammar should be extensively improved. There are some comments, as follows.
1. In the “Evolution of MRI on assessment of PCLs” section, “…isotropic image acquisition that offer the highest image resolution…” should be revised as “…isotropic image acquisition that offers the highest image resolution…”.
2. In the “Prevalence of PCLs in MRI” section, in the sentence “The effect of technological improvements in PCL prevalence, was illustrated…”, “,” should be removed.
3. In the “Prevalence of PCLs in MRI” section, the numerical format of the prevalence in Table 1 should be consistent. The number “49.1%” should be revised as “45.9%”. At the same time, please give the reasons for the large variations in PLC prevalence among the observational studies.
4. In the “MRI features of PCLs” section, the full name of “IPMN” should be given.
5. In the “Role of MRI to diagnose specific cyst types” section, “…so cyst diagnosis in clinical practice heavily relies in proper imaging interpretation…” should be revised as “…so cyst diagnosis in clinical practice heavily relies on proper imaging interpretation…”.
6. In the “Role of MRI to diagnose specific cyst types” section, the role of MRI to diagnose retention cyst should be described in more details.
7. In the “Role of MRI to diagnose specific cyst types” section, “…main-duct and mixed-duct IPMNs as compared to BD-IPMN” should be revised as “…main-duct and mixed-duct IPMNs as compared to branch-duct IPMNs”. Meanwhile, the expression of “branch-duct IPMN” and “BD-IPMN” should be consistent.
8. In the “Role of MRI to diagnose specific cyst types” section, the value of “[AUC] 9.95” is wrong.
9. In the “Role of MRI to diagnose specific cyst types” section, “… a small branch-duct IPMNs…” should be revised as “… a small branch-duct IPMN…”.
10. In the “Role of MRI to diagnose specific cyst types” section, “None of these morphologic features is specific of IPMNs…” should be revised as “None of these morphologic features are specific to IPMNs…”.
11. In the “Role of MRI to diagnose specific cyst types” section, “… fibrotic changes noticed on histology” should be revised as “… fibrotic changes noticed in histology”.
12. In the “Role of MRI to diagnose specific cyst types” section, “… at increased risk to develop SCAs” should be revised as “… at increased risk of developing SCAs”.
13. In the “Role of MRI to diagnose specific cyst types” section, “…and have thin septations” should be revised as “…and has thin septations”.
14. In the “Role of MRI to diagnose specific cyst types” section, “… that would differentiate it from the other cystic lesions” should be revised as “… that would differentiate them from the other cystic lesions”.
15. In the “Role of MRI to diagnose specific cyst types” section, the full name of “EUS” should be given.
16. In the “Role of MRI in detecting advanced neoplasia” section, in the sentence “Several society guidelines have proposed classification systems to risk stratify the malignant potential of mucinous lesions based on the presence of high-risk or worrisome imaging features”, references should be added.
17. In the “Role of MRI in detecting advanced neoplasia” section, the full names of “AGA”, “ACG”, and “ACR” should be given.
18. In the “Methods to improve evaluation of PCLs with MRI” section, “OR 1.28” should be revised as “OR=1.28”.
19. In the “Methods to improve evaluation of PCLs with MRI” section, “…the addition of secretin did not significantly increased ductal communication…” should be revised as “…the addition of secretin did not significantly increase ductal communication…”.
20. In the “Methods to improve evaluation of PCLs with MRI” section, the full names of “IV”, “ADC”, and “HGD” should be given.
21. In the “Methods to improve evaluation of PCLs with MRI” section, “…also kwon as quantitative imaging …” should be revised as “…also known as quantitative imaging …”.
22. In figure legends, “demonstrate” should be revised as “demonstrates”.
23. In Figure 6, “Macrocytic SCA” should be revised as “Macrocystic SCA”, and “9” should be revised as “(”.
Author Response
Thank you for the comments. We have responded to each of the queries. Please find our responses below.
Reviewer 2
- In the “Evolution of MRI on assessment of PCLs” section, “…isotropic image acquisition that offer the highest image resolution…” should be revised as “…isotropic image acquisition that offers the highest image resolution…”.
- Thank you. This has been corrected
- In the “Prevalence of PCLs in MRI” section, in the sentence “The effect of technological improvements in PCL prevalence, was illustrated…”, “,” should be removed.
- The “,” has been removed as requested.
- In the “Prevalence of PCLs in MRI” section, the numerical format of the prevalence in Table 1 should be consistent. The number “49.1%” should be revised as “45.9%”. At the same time, please give the reasons for the large variations in PLC prevalence among the observational studies.
- Thank you for the comment. The numerical format has been edited to be consistent. The number requested has been corrected. We have made more clear the explanations that may explain the large variations of PCL prevalence among the observational studies.
- In the “MRI features of PCLs” section, the full name of “IPMN” should be given.
- Addressed, thank you
- In the “Role of MRI to diagnose specific cyst types” section, “…so cyst diagnosis in clinical practice heavily relies in proper imaging interpretation…” should be revised as “…so cyst diagnosis in clinical practice heavily relies on proper imaging interpretation…”.
- Thank you. This has been revised.
- In the “Role of MRI to diagnose specific cyst types” section, the role of MRI to diagnose retention cyst should be described in more details.
Thank you for the comment. This has been included in the manuscript. “Pancreatic retention cysts are rare benign lesions that can be found in the setting of chronic pancreatitis, cystic fibrosis, or pancreatic tumors. Mechanistically, these cysts represent a dilated segment of the pancreatic duct due to focal duct obstruction from strictures, mucin plugs, calculi, or tumors. The radiologic appearance of retention cysts has not been widely described and it may be difficult to diagnose these cysts with MRI alone as they share characteristics with other PCLs. In a small study of 16 patients with retention cysts, these lesions were well-defined, round, without solid component hypointense on T1WI images, and hyperintense on T2WI images. These lesions may have thin septations, communication to the main pancreatic duct, and ductal dilation.”
- In the “Role of MRI to diagnose specific cyst types” section, “…main-duct and mixed-duct IPMNs as compared to BD-IPMN” should be revised as “…main-duct and mixed-duct IPMNs as compared to branch-duct IPMNs”. Meanwhile, the expression of “branch-duct IPMN” and “BD-IPMN” should be consistent.
This has been edited as requested.
- In the “Role of MRI to diagnose specific cyst types” section, the value of “[AUC] 9.95” is wrong.
We appreciate the reviewer’s comment. The AUC value has been changed to 0.95.
- In the “Role of MRI to diagnose specific cyst types” section, “… a small branch-duct IPMNs…” should be revised as “… a small branch-duct IPMN…”.
Thanks. This has been revised.
- In the “Role of MRI to diagnose specific cyst types” section, “None of these morphologic features is specific of IPMNs…” should be revised as “None of these morphologic features are specific to IPMNs…”.
Thanks. “Is” has been changed for “are”
- In the “Role of MRI to diagnose specific cyst types” section, “… fibrotic changes noticed on histology” should be revised as “… fibrotic changes noticed in histology”.
This has been revised.
- In the “Role of MRI to diagnose specific cyst types” section, “… at increased risk to develop SCAs” should be revised as “… at increased risk of developing SCAs”.
Addressed, thank you.
- In the “Role of MRI to diagnose specific cyst types” section, “…and have thin septations” should be revised as “…and has thin septations”.
This has been corrected.
- In the “Role of MRI to diagnose specific cyst types” section, “… that would differentiate it from the other cystic lesions” should be revised as “… that would differentiate them from the other cystic lesions”.
Revised.
- In the “Role of MRI to diagnose specific cyst types” section, the full name of “EUS” should be given.
Thanks. Addressed.
- In the “Role of MRI in detecting advanced neoplasia” section, in the sentence “Several society guidelines have proposed classification systems to risk stratify the malignant potential of mucinous lesions based on the presence of high-risk or worrisome imaging features”, references should be added.
Thank you. References have been added.
- In the “Role of MRI in detecting advanced neoplasia” section, the full names of “AGA”, “ACG”, and “ACR” should be given.
This has been provided.
- In the “Methods to improve evaluation of PCLs with MRI” section, “OR 1.28” should be revised as “OR=1.28”.
This has been revised as suggested by the reviewer.
- In the “Methods to improve evaluation of PCLs with MRI” section, “…the addition of secretin did not significantly increased ductal communication…” should be revised as “…the addition of secretin did not significantly increase ductal communication…”.
Addressed, thank you.
- In the “Methods to improve evaluation of PCLs with MRI” section, the full names of “IV”, “ADC”, and “HGD” should be given.
Thank you. This has been provided.
- In the “Methods to improve evaluation of PCLs with MRI” section, “…also kwon as quantitative imaging …” should be revised as “…also known as quantitative imaging …”.
Corrected.
- In figure legends, “demonstrate” should be revised as “demonstrates”.
This has been revised.
- In Figure 6, “Macrocytic SCA” should be revised as “Macrocystic SCA”, and “9” should be revised as “(”.
Addressed, thank you.
Round 2
Reviewer 1 Report
The article has improved with the changes made in the manuscript.
Reviewer 2 Report
Language should be further improved.